# Incentives behind and Experiences of Being Active in Working Life after Age 65 in Sweden

**DOI:** 10.3390/ijerph192315490

**Published:** 2022-11-22

**Authors:** Marie Bjuhr, Maria Engström, Anna-Karin Welmer, Magnus Lindberg, Britt-Marie Sjölund

**Affiliations:** 1Faculty of Health and Occupational Studies, Department of Caring Sciences, University of Gävle, 801 76 Gävle, Sweden; 2Aging Research Center & Division of Physiotherapy, Department of Neurobiology, Care Sciences and Society, Karolinska Institute, 171 65 Stockholm, Sweden

**Keywords:** extended working life, multidimensional health, older workers, qualitative content analysis

## Abstract

Since individual and societal expectations regarding the possibility of an extended working life after the expected retirement age are increasing, research on sustainable working life combined with healthy ageing is needed. This study explores the incentives behind and experiences of an extended working life after the expected retirement age of 65 among Swedish people. The inductive qualitative content analyses are based on 18 individual semi-structured interviews among persons 67–90 years old with varying characteristics and varying experiences of extended working lives. The analyses revealed that working contributed to (1) sustained internal resources, i.e., cognitive function, physical ability and increased vigor; (2) sustained external resources, i.e., social enrichment, better daily routines and economic benefits; (3) added meaningfulness to life, i.e., being needed, capability and satisfaction with working tasks. Meanwhile, having flexible working conditions enabled a satisfying balance between work and leisure. Altogether, these different aspects of overall health and working life were interpreted as contributing to increased feelings of vitality, the innermost dimension of health. Conclusions: regardless of biological age, our results indicate that being able to remain active in working life can be beneficial to vitality and could make these results valuable for both health-care personnel and employers.

## 1. Introduction

In recent decades, there has been an ongoing increase in the proportion of older people in the population with longer life expectancies and an increased number of healthy years [1]. Individual [2] and societal expectations [3] regarding the possibility of an extended working life is expanding. Since in most western countries, senior workers have increased their participation in working life [3,4], research and actions focusing on sustainable working life in combination with healthy aging are needed [5].

Known factors associated with being active in working life after the expected retirement age include a higher educational level, working within highly skilled occupations/professions, good health status, and being male [6,7,8,9]. Qualitative studies have revealed that perquisites and motivations for working life after the expected retirement age can include work satisfaction and good organizational support [10,11,12,13,14,15], financial incentives [13,14,15,16,17], social benefits [11,15,16], and/or having flexibility in work, e.g., time or tasks [14,15,16,18]. Earlier research has also shown that both managers [15,19] and workers [15,20] highlight good health as a key facilitator to an extended working life.

Despite the amount of research that focuses on working life after the expected retirement age, there is still a lack of studies embracing the incentives behind and experiences of extended working lives among persons 65 years or older from varying occupations. In this study, the definition of the expected retirement age is 65 years. In Sweden, the flexible statutory pension age is 62–68 years. However, in consultation with the employer, it is possible to work longer. In 2021, the average exit age from work in Sweden was 64.2 years [21]. Typically, in earlier research regarding working life after the expected retirement age, which mostly used a quantitative approach, health was defined by the number of diagnosed diseases, self-reported health, and/or functional disability [22]. Eriksson’s (1994) theory of multidimensional health embraces different levels (the human being as an entity of body, soul, and spirit) and dimensions (doing, being and becoming) of an individual’s overall health [23] and can utilize extended knowledge about the complexities surrounding older peoples’ participation in working life. This theory represents a non-medical paradigm within nursing and caring sciences from a human science point of view [24]. Eriksson discovered and illustrated that different dimensions of health and the pure concept of health means wholeness and holiness. This pluralistic view of the human being is the foundation of the proposition regarding the technology of health (i.e., human health behaviors; the level of doing) and the ontology of health (i.e., a deeper understanding; the levels of being and becoming) [25]. By raising awareness of the different dimensions of health, it is possible to avoid health being limited to behaviors or good health habits. Another proposition in the theory of multidimensional health is that the experience of health involves a constant movement between the different dimensions. In this movement, the individual can grow and become what she or he is meant to be [23,25]. Therefore, the purpose of this study was to explore the incentives behind and experiences of an extended working life after the expected retirement age of 65 among Swedish people. Furthermore, using a qualitative approach the results of this study are discussed in relation to the theory of multidimensional health.

## 2. Materials and Methods

### 2.1. Design

This is a descriptive and interpretative qualitative study that applied an inductive approach.

### 2.2. Participants and Data Collection

The inclusion criterion that resulted in participants aged between 67 and 90 years was people who had been active in working life for at least one year beyond the age of 65. The 18 participants were recruited in connection with a follow-up assessment associated with their participation in the Swedish National Study on Ageing and Care (SNAC-project) [26]. Thirteen people were selected through purposive sampling and five people through snowball sampling, i.e., purposive participants identified through contacts with the already selected participants or the researchers responsible for the project. The purposive sampling aimed to yield participants with a variety of characteristics based on known factors associated with being active in working life after the expected retirement age. The participants’ characteristics are presented in Table 1.

The interviews were conducted between August 2021 and February 2022. The interviews, per the participant’s choice, were face-to-face at an undisturbed place (n = 5), by telephone (n = 10), or via video conferencing (n = 3). The duration of the interviews varied between 14 and 37 min. A semi-structured interview guide was used. Initially, the participants were asked to give oral descriptions regarding background data. They were asked about their level of education and their current living circumstances, i.e., if they were living alone or cohabiting and if they were living in an urban setting (densely populated environments that provides basic facilities for human activity) or in a rural setting (countryside areas or villages with a low-density population) [27]. They were asked to describe their overall health (in this study referred to as self-reported health) and if they had any diagnosed diseases. The participants were also asked about their professional life and their current occupational status, i.e., what percentage of time they were employed based on a full-time 40 h work week. The first question, which was related directly to the study’s aim, was directed toward the incentives behind the decision to work beyond the age of 65. The following questions focused on experiences of working life; see Appendix A. The interviews were audio-recorded and transcribed into a text document.

### 2.3. Analysis

Qualitative inquiry is commonly oriented toward exploration and inductive logic, and content analysis can be defined as a general term for organizing and categorizing the content of a narrative text [28]. In this study, the transcribed interviews were analyzed with inductive qualitative content analysis as suggested by Graneheim and Lundman (2004), and Lindgren, Lundman and Graneheim (2020) [29,30]. This method clarifies the analysis steps that enable a qualitative content analysis to be both descriptive and interpretative, revealing both the depth and meaning in the participants’ utterances [31].

Initially, the first and the last authors read the text from all of the interviews several times to determine an overall sense of the whole. In the next step, based on the aim, the text was divided into identified meaning units, which were condensed without losing the essential meaning. The meaning units were then labeled with codes, and these codes were sorted into different groups based on similarities and differences to form categories. This initial step was performed by the first author. The labeling and contents within the categories were discussed together with the last author several times. These categories, which have a low degree of abstraction, show the manifest content of the text. The aim was to describe the participants’ incentives and experiences and “what” comprises the similarities within each category. In the next step, categories with common content were brought together into four different subthemes. The headings of these subthemes have a higher level of abstraction and are more distant from the original text. Finally, one overall theme could be interpreted as a “red thread” in all categories and subthemes. The theme highlights “how” the incentives behind being active in working life after the expected retirement age is described, an interpretation of the latent content of the text. During the entire process, the focus shifted back and forth between the identified categories and subthemes and the whole text. Examples of the steps in the analysis process are shown in Appendix A.

### 2.4. Ethical Considerations

The study was approved by the Swedish Ethical Review Authority (D.nr. 2021-04236). Verbal and written information about the project was provided prior to participation and the signing of written informed consent. Participation was voluntary, and participants could withdraw their participation at any time without explanation. Confidentiality was guaranteed and the interviews were assigned a code (I.1–I.18)

## 3. Results

### 3.1. Working Increases Feelings of Vitality

An overview of the findings with the included theme, subthemes, and categories is shown in Table 2. Being active in working life after 65 contributed to feelings of vitality, which was interpreted as one consistent theme from the participants’ experiences. Incentives to remain active in working life at an older age were described as life-affirming, despite the knowledge that it could end at any time. Working meant to live in the present, to remain open to new impressions, and a longing to continue to grow and develop throughout life. Sustained internal and external resources, added meaningfulness, and flexible working conditions contributed to the feeling of vitality caused by work increase.

*You can’t sit and wait for life to end, but you know it’s coming, but not when, but you have to keep living and yes, try to live in the present and try to be open to new ideas, that’s what I think*.[I:2]

### 3.2. Sustained External Resources

The participants claimed that working life maintained their external resources since it was socially enriching, contributed to better daily routines, and was economically beneficial. The participants described the incentives for being active in working life as including the experiences of fellowship and experiencing a social context with other people at work through collaboration with workmates and meetings with customers, patients, passengers, or children the participants met during work. This socially enriching context also included break times at work and their everyday talks and discussions with colleagues.

*Colleagues and children recognize me; it makes it more comfortable easier and more fun to go to work. It’s not just a job it’s a fellowship. Everybody knows each other*.[I:18]

Another socially-enriching aspect was the increased opportunity to meet younger people. Outside of work, some of the participants maintained that they mainly only met others their own age, except for family members.

The participants described how working life contributed to an improved structure in their daily routines and circadian rhythms. This was also beneficial in regard to other aspects of everyday life, e.g., house cleaning, grocery shopping, or cooking, and therefore contributed to their well-being. Maintained daily routines were also described as counteracting passivity such as just staying home, waking up late, having nothing planned, and only watching TV.

*Work helps to maintain structure in life, you get up, eat breakfast, go to work and so on. It is important to me*.[I:9]

Another motivation for being active in working life was expressed financially as economically beneficial. The participants described that because they had a meager pension, their wages from work were valuable. Being able to maintain one’s standard of living and being able to, e.g., travel, eat better food, or save money for their grandchildren was something that strengthened the participants’ well-being in daily life.

*I’ve never had a high-paying job, therefore, I don’t have a large pension income, but I have a healthy body, so I can work and earn some money. This means that I can save some money, but also be able to afford to do nice things sometimes e.g., travel. It is nice not to have to worry about money all the time*.[I.4]

### 3.3. Sustained Internal Resources

The participants explained that working life maintained their internal resources since it helped to keep their minds active, promoted physical activity, and increased their vigor.

Working contributed to the preservation of their cognitive capacity by keeping their minds active, which was expressed as performing various work tasks, learning new things, and staying informed about current events in the world. Experiences such as learning new computer systems or performing tasks that require cognitive ability were described as health-promoting and a motivation to remain in working life. The participants asserted that keeping the mind active possibly reduces the risk of dementia and staying cognitively “well preserved”.

*All my sisters have been diagnosed with dementia, so yes, I am a little worried. But I think as long as you keep your brain active, I think it (having dementia) might be delayed*.[I:13]

Working life also benefited physical health, which they said occurred when cycling or walking to work or the work itself that required physical activity.

*This is a physical job, where you have to exert yourself. But that’s why the body stays in pretty good shape*.[I:8]

Additionally, experiences of coping with physical burdens from work contributed to physical activities outside of work. Having an operation on an arthritic knee to make it possible to continue being active in working life is an example of how working life could indirectly be beneficial to physical health in life overall.

Working life also increased vigor since the participants described how it increased joy and inner energy that also benefitted their leisure time. From working life, the participants had the possibility to stay active, which also counteracted their fear of being active in other contexts such as having the courage to meet new people or continuing to ride a bicycle.

*It’s good for your health, the job keeps you active and you continue to do things, so you don’t get frightened of everything either*.[I:1]

### 3.4. Added Meaningfulness

Being active in working life added meaningfulness to the participants’ daily lives. This was expressed by their comments of feeling good with the knowledge that they were still needed for their professional/work competence, that they experienced work satisfaction, and that working promoted feelings of being capable.

The participants felt good to still be needed in working life, and that their employer and/or customers valued their experience and competence. Some described how working in an occupation that is experiencing a staff shortage gave them a sense of meaningfulness since they could fill that need. Their contribution improved society, which added to their sense of security and their decision to remain active in working life. The participants felt privileged because they had acquaintances who were not allowed to continue to be active in working life even though they wanted to.

*I think many people get depressed when they are forced to stop working; to hear there is no more work for you*.[I:6]

The participants also described that they felt genuine job satisfaction with their work tasks.

*I like to work on cars, it was a boyhood dream to have my own auto repair shop*.[I:8]

If the work was considered pleasurable, the participants felt their age had no impact on whether they would continue to be active in working life. Furthermore, the work also added a sense of meaning as it encouraged creativity and was considered stimulating, with new challenges.

A motivation to remain active in working life was expressed through their described feelings of being capable, which brought well-being. Being able to support younger coworkers with their experience and skills, achieving good results, and being able to cope with challenging tasks are a few examples.

### 3.5. Having Flexible Working Conditions

New flexible working conditions where the participants themselves had influence over when, what tasks, and to what extent they would work were other incentives for being active in working life.

The participants expressed that having the opportunity to choose tasks gave them the possibility to refrain from tasks that felt burdensome or stressful during their previous work life and enabled a new, more positive, perception of work.

*I can decide for myself when I want to work and with jobs that feel fun. Being a welder can be quite heavy work, so it’s good I can choose*.[I:14]

The new working conditions also meant a better balance between work, recovery, and leisure, which is why the participants described the experience of being active in working life as beneficial to their overall health. The participants claimed that if they were still active in working life, they felt a sense of freedom as well.

## 4. Discussion

In summary, our findings suggest that the incentives behind and experiences of being active in working life after age 65 involve different aspects of overall health. Working contributed to sustained internal and external resources and added meaningfulness to life, which are important prerequisites for health and well-being. Having flexible working conditions enabled a satisfying balance between work and leisure. Altogether, these different aspects of overall health and working life contributed to increased feelings of vitality, the innermost dimension of health.

Our findings regarding how working can contribute to increased feelings of vitality are in line with previous research that suggests that strengthened vitality in work goes hand in hand with increased internal and external health resources through work [11]. According to the dimension of “becoming” in the multidimensional health theory, health can be described as a force to energize life, joy, and a willingness to live and grow [23]. Meaningful activities and inner strength have been described as important sources of vitality among older individuals [32], and the participants in this study considered working to be energizing and described how working sustained physical and mental resources and health.

Sustained internal and external resources can be related to the multidimensional health dimension of “doing”, which means finding a life practice, and is expressed in good health habits. Internal resources are defined as, e.g., physical, mental, and spiritual elements of an individual’s prerequisites for experiencing health and well-being, while external resources are, e.g., relationships with family members or social networks at work [33]. Additionally, our findings revealed that experiences of being active in working life after the age of 65 added a sense of meaningfulness. Research targeting retirement preferences also showed that meaningful work was a motive for wanting to retire later in life [10,14]. Based on the multidimensional health theory, the “being” dimension embraces the importance of experiencing meaningful communities when individuals are striving for well-being and harmony [23]. As in previous research [18], our findings indicate that new working conditions with increased flexibility were among the incentives for staying active in working life. Additionally, flexibility in work tasks enabled the participants to have a new, more positive, perception of work. It has also been found that individuals who worked beyond the expected retirement age, with reduced working hours, reported significant improvements in their job quality [34].

In this study, none of the participants considered that they had been forced, due to economic reasons, to remain active in working life. However, it is important to consider that some people may have negative experiences of extended working life since, despite health problems, they still need income from work [35]. According to the World Health Organization, the first key consideration of healthy ageing is diversity, i.e., a heterogeneous view of the aging population. The second is inequity since diversity in an individual’s capacity is largely effected by the cumulative advantages and disadvantages experienced during their life [36]. This is also shown in previous research, i.e., people with a low level of education and low income have an 80% higher risk of working after the expected retirement age for financial reasons vs. other reasons [37]. Therefore, it is essential that we do not make general assumptions regarding working at an older age and how it may benefit health and well-being. Decision makers and employers need to adopt actions that reduce the unequal conditions for extended working life so it can be favorable for healthy ageing. Our findings suggested that flexible working conditions enabled an extended working life even among people with physically demanding jobs. Future research on this topic is needed, especially directed towards the question of how extending working life after vs. exiting from working life before the expected retirement age interacts with one’s overall health, i.e., from a multidimensional health perspective.

### Trustworthiness

The aim was to describe various experiences, which is why the participants needed to have varying characteristics. Table 1 can facilitate an evaluation of the sample [29]. One limitation regarding transferability is that none of the included participants considered that they were “forced” to remain active in working life due to, e.g., financial reasons. However, many of the participants described economic benefits as a motivating aspect of remaining active in working life due to their meager pensions. The participants were invited to participate in this study in connection with a follow-up assessment associated with their involvement in the SNAC project, i.e., an opportunity sampling. This method could be a limitation that affects the transferability, since one disadvantage is that a researcher is likely to select people like themselves, or those with whom they have a good contact. However, there were two research nurses at two different SNAC research centers, one located in a rural setting and one in an urban setting, who were involved in the recruiting of participants for our study. Only one of those nurses (B-M. S) is involved in this manuscript. Another limitation might be that 5 of the 18 participants were only active 20–35% in working life. This is mostly explained by the relatively high mean age in this study. However, during the interviews, those participants revealed that they were fully employed up to age 67 years. To enable judgment of the data collection, we offer the transparency of the interview guide (Appendix A). All interviews were conducted by the first author (a registered nurse and Ph.D. student). After the first five interviews, with the aim of strengthening the quality of the upcoming interviews and the credibility of the data collection, the data were discussed with the entire research group. The amount of data gathered after the completion of all the interviews was considered adequate to capture the significant variations in the participants’ experiences [31]. Despite the limited duration of some of the interviews, it was determined that they offered sufficiently rich material considering the aim. All co-authors have previous health science experience in research, education, and practice. During the analysis process, the co-authors discussed the identified categories and subthemes until a consensus was reached. One challenge in qualitative content analysis is whether categories, subthemes, and themes maintain a credible level of abstraction and interpretation to facilitate the assessment of the credibility and dependability of the data analysis [30], which is why we submitted examples of the steps in the process (Appendix A). The findings are presented in Table 2, and in text with relevant quotes.

## 5. Conclusions

Our study extends the knowledge regarding the participants’ perceptions of how remaining active in working life can be beneficial to health. A multidimensional perspective of health enabled a discussion about experiences of extended working life and an individual’s overall health. These findings are of value to health-care personnel when meeting older individuals within different health-care contexts. Regardless of biological age, our results indicate that being able to remain active in working life can be beneficial to vitality, the innermost dimension of health [25]. Actions to provide flexible working conditions, especially for disadvantaged senior workers, may facilitate working life after the expected retirement age with extended equality. However, further knowledge regarding senior workers and their experiences and of the possibilities to achieve healthy ageing in relation to their occupational status is still needed [5].

## Figures and Tables

**Table 1 ijerph-19-15490-t001:** The participants’ characteristics.

Characteristics	N = 18	Mean	Median
**Age**		74.4	74.5
67	1		
68	3		
69	4		
72	1		
73	1		
76	1		
77	1		
78	4		
79	1		
82	1		
90	1		
**Sex**			
Female	11		
Male	7		
**Living situation**			
Urban setting	11		
Rural setting	7		
Living alone	8		
Cohabitating	10		
**Health Status ^**			
Overall health	18		
Good			
Diagnosed disease(s) *			
None	4		
Hypertension	7		
Cardiovascular	2		
Musculoskeletal	4		
Cancer	2		
Asthma	1		
Blood disease	1		
**Education**			
Elementary	4		
High school	3		
Vocational school	3		
University	8		
**Currently employed (%)**		59.2	55
20–35	5		
40–70	7		
75–120	6		
**Occupation ^#^**			
Health care	2		
Driver	2		
Education/teaching	3		
Hairdresser	2		
Illustrator	2		
Shop manager	1		
Community tourism manager	1		
Carpenter	1		
Mechanic	1		
Welder	1		
Architect	1		
Pharmacist	1		

^ Participants own description of overall health and diagnosed disease. * Some participants declared more than one disease. ^#^ The percentage the participants participated in full-time employment based on a 40-h work week.

**Table 2 ijerph-19-15490-t002:** Overview of findings: theme, subthemes, and categories.

Theme	Working Increases Feelings of Vitality			
Sub-theme	Sustained external resources	Sustained internal resources	Added meaningfulness	Flexible working conditions
Category	Socially enriching	Good for my cognitive function	I am still needed	I decide for myself what
	Maintained everyday routines	Good for my physical ability	Work satisfaction	I decide for myself when
	Economically beneficial	Increases my vigor	I am still capable	I decide for myself how much

## Data Availability

The data presented in this study are not publicly available, as the participants were not asked to provide permission for the research team to share their data. Detailed analyses of the confidential transcripts are available on request from the corresponding author.

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
