# Peer review of "Incentives behind and Experiences of Being Active in Working Life after Age 65 in Sweden"

_ijerph, 2022, doi:10.3390/ijerph192315490_

Round 1

Reviewer 1 Report

The main objective of this study was to explore the incentives behind and experiences from an extended working life after the expected retirement age of among Swedish people.

Some considerations are recommended and proposed in case they can provide greater quality to the document.

1. A complete revision of the text is recommended to correct typographical errors (especially regarding line endings, e.g. lines 32, 50, 129, 130, etc.)

2. It is recommended to include in the abstract a more detailed description of the method used, for a better and more precise identification of how the research was conducted.

3. In relation to the characterization of the sample, greater precision in many elements would be desirable (besides, n=18 could be lower than desirable for this type of study):

A) Why were these different age ranges used? (67-72, 73-78 and 79-90) Is there any justification? even, being a small sample, the age of each participant could be reported.

B) What does rural and urban mean? is it by number of inhabitants? how many?

C) Self-reported health: here a simple questionnaire should have been applied to better specify the data. Not having been so, what does "GOOD" mean? what scale was used? good, medium, bad? What aspects was the participant's opinion based on?

D) It is assumed that the diseases were officially diagnosed, but the source should be reported (family doctor, hospital, company medical examination, etc.)

E) Please, it is necessary to explain what the percentages referring to "currently employed" mean in both columns.

F) In general, and by not performing a multifactorial analysis between the characteristics of the sample and the results obtained, the reason for the characterization of the sample in the selected items is not very well understood.

4. In the analysis section, was it based on a method already used and previously validated in previous research, or has it been implemented for the first time in this research?

5. In the paragraph dedicated to the Limitations, please clearly detail the limitations, and what is related to the "procedure" should be in the Method section. 

Reviewer 2 Report

Bjuhr_Being active 65 years old_IJERPH_2022

I commend the authors on the completion of this manuscript. I really liked the theme and design. Overall it is well designed and on an innovative and important topic. I have a few concerns highlighted below.

Abstract

Conclusions. Please add some reference to the aspect related to “senior workers and their experiences of the possibilities to achieve healthy ageing in relation to their occupational status.”

1. Introduction

Line 52. Please add a little more information about: Katie Eriksson's caring theories.

Line 55. Please, include reference before comma. 

2. Materials and Methods

Table 1. Please clarify the variable “currently employed”. What is the percentage based unit. For example: The percentage is on 40 hours week? The percentage refers to what?

4. Discussion

Trustworthiness Line 296. Please add a limitation regarding the opportunity sampling method affecting transferability. 

Round 2

Reviewer 1 Report

Thank you very much for your answers, which have clarified some aspects of the manuscript, however, I still have some reservations about how the study has been conducted.

Even so, I consider the topic very interesting, which will require continuity in its investigation; while I encourage and recommend a greater scientific guarantee, more appropriate statistical methods and analyzes so that the conclusions can be used to intervene.
